# Development of Dog Immune System: From *in Uterus* to Elderly

**DOI:** 10.3390/vetsci6040083

**Published:** 2019-10-21

**Authors:** Maria Pereira, Ana Valério-Bolas, Cátia Saraiva-Marques, Graça Alexandre-Pires, Isabel Pereira da Fonseca, Gabriela Santos-Gomes

**Affiliations:** 1Global Health and Tropical Medicine (GHTM), Instituto de Higiene e Medicina Tropical (IHMT), Universidade Nova de Lisboa (UNL), R. da Junqueira 100, 1349-008 Lisboa, Portugal; pereirabia@hotmail.com (M.P.); anasbolas@gmail.com (A.V.-B.); 2Instituto Politécnico de Portalegre (IPP), Praça do Município 11, 7300-110 Portalegre, Portugal; 3Agrarian School of the Polytechnic Institute of Viseu, Quinta da Alagoa—Estrada de Nelas Ranhados, 3500-606 Viseu, Portugal; 4Centro Interdisciplinar de Investigação em Sanidade Animal (CIISA), Faculdade de Medicina Veterinária (FMV), Universidade de Lisboa (UL), Av. Universidade Técnica, 1300-477 Lisboa, Portugal; msfcatia@gmail.com (C.S.-M.); gpires@fmv.ulisboa.pt (G.A.-P.); ifonseca@fmv.ulisboa.pt (I.P.d.F.)

**Keywords:** dog, immune system, immunity, passive immune transfer, immunity development

## Abstract

Immune system recognize and fight back foreign microorganisms and inner modifications that lead to deficient cell and tissue functions. During a dog’s life, the immune system needs to adapt to different physiological conditions, assuring surveillance and protection in a careful and controlled way. Pregnancy alters normal homeostasis, requiring a balance between immunity and tolerance. The embryos and fetus should be protected from infections, while the female dog must tolerate the growing of semi-allografts in her uterus. After birth, newborn puppies are at great risk of developing infectious diseases, because their immune system is in development and immune memory is absent. Passive transfer of immunity through colostrum is fundamental for puppy survival in the first weeks of life, but hampers the development of an active immune response to vaccination. At the end of life, dogs experience a decline in the structure and functional competence of the immune system, compromising the immune responses to novel antigenic challenges, such as infections and vaccines. Therefore, the current article reviews the general processes related to the development of the dog´s immune system, providing an overview of immune activity throughout the dog’s life and its implications in canine health, and highlighting priority research goals.

## 1. Introduction

Information about the development of the dog’s immune system is limited, dispersed, and some older studies are difficult to access. The immature innate and adaptive immune system of the newborn puppy matures and acquires memory while the animal grows up, and finally declines in the latter stages of life. These changes in the immune system bring different challenges throughout the dog’s life and predispose the animal to different types of infections, immune-mediated diseases, and neoplasms. Thus, this paper aims to summarize, in a simple and understandable way, some aspects of the development of the immune system with implications in the dog’s health throughout its life, helping veterinarians to delineate prophylactic and therapeutic approaches.

This paper starts with a brief overview of the immune system. Some development aspects of the dog´s immune system will be presented from a chronological point of view, starting with the immunological characterization of canine pregnancy and maternal transference of immunity. Then, it will be addressed the maturation of primary and secondary immune organs and of the immune response, and the immunosenescense establishment. Finally, the concluding remarks and future trends will be presented, focusing on some aspects that deserve better study.

## 2. Immune System Overview

The immune system determines the survival of the individual, discriminating self from non-self. The organism uses multiple defense mechanisms, simultaneously, to ensure the absence of disease. Physical barriers prevent the penetration of invading microorganisms, and the innate immune system has mechanisms of rapid response, such as inflammation, complement system, and antimicrobial molecules, to prevent infection. Neutrophils and macrophages play an important role as innate immune cells [1].

Neutrophils are the most abundant white blood cells, comprising up to 75% of the total leukocyte count in the adult dog. These relatively short living cells constitute the primary defense against microbial infections. Although neutrophils leave the bone marrow pre-equipped with cytoplasmic granules, containing a variety of antimicrobial molecules, they circulate in the blood stream as dormant cells. When properly stimulated, neutrophils migrate to tissues where they set into action a variety of mechanisms able to contain the infection, namely phagocytosis, release of neutrophil extracellular traps (NETs), and exocytosis of granular molecules [2,3,4]. 

Blood monocytes comprises up to 5% of the total leukocyte count in the adult dog. Monocytes migrate from the bloodstream into the tissues, acquiring specific phenotypes and functional characteristics. These cells can perform diverse activities, such as phagocytosis, the release of macrophage extracellular traps (METs), antigen presentation, tissue repair, and also function as scavenger cells [5,6,7].

Neutrophils and macrophages use pre-existing receptors, such as pattern recognition receptors (PRRs) to recognize molecular antigenic patterns (PAMPs) shared by several microorganisms, allowing them to phagocyte and destroy invaders. Natural killer (NK) cells have receptors for surface molecules expressed by normal cells. When these molecules changed or are absent in infected and altered cells, NK cells can induce cytolysis or apoptosis of target cells [1].

Antigen presenting cells (APCs) establish the link between innate and adaptive immune response. APCs phagocyte microorganisms, digest them into small antigenic fragments and expose them on the cell surface in association with major histocompatibility complex (MHC) molecules. Antigens presenting by APCs can be recognized by lymphocytes. Dendritic cells are sentinel cells present in almost the entire body, and potent APCs, capable of stimulating naïve T cells. Macrophages and B lymphocytes are less potent APCs [1].

Once lymphocytes have left the central lymphoid tissues (bone marrow, thymus), they are carried by the blood to the peripheral lymphoid tissues. Up to 50% of peripheral blood leukocytes are lymphocytes. Peripheral lymphoid tissues have a highly organized architecture, with distinct areas of B cells and T cells. The survival of newly formed lymphocytes is determined by their interactions with other cell types. If a lymphocyte does not encounter its specific antigen (that is presented by APCs) and become activated, it dies. Properly activated lymphocytes are long-living cells [8,9].

The adaptive immune response is extremely efficient, because it responds specifically to the invading microorganism and generates immune memory, but it takes longer to be activated. Antigen presentation by APCs to lymphocytes occurs in the lymphoid organs, triggering the adaptive immune response. The lymphocytes have specific receptors to recognize antigens presented by APCs. These receptors are generated randomly, directing the generation of a highly diverse lymphocyte population capable of recognizing virtually all foreign antigens that the animal may encounter. T cell antigen receptor (TCR) and B cell antigen receptors (BCR) are complex structures formed by various proteins. According to polypeptide chains of the TCR, T lymphocytes can be classified as αβ or γδ. T lymphocytes express on its surface the signal transduction complex named cluster of differentiation (CD)3, that is connected to TCR and transmit the signal to the cell when antigen binding occurs. Associated with TCR, αβ-T cells present one of two molecules, CD4 or CD8. The type of molecule allows distinguish T cells into two subpopulations: T helper (Th) (CD4^+^) and T cytotoxic (CD8^+^). CD4 molecules on Th cells ligate MHC class II molecules used by APCs to present exogenous antigens. CD8 molecules on T cytotoxic lymphocytes bind MHC class I molecules that are used by infected or abnormal nucleated cells to present endogenous antigens [1,10]. MHC class I molecules are present in all nucleated cells, while MHC class II molecules are expressed only by APCs. “Unconventional” γδ-T cells are present in the spleen and mucosal surfaces, and play a role in regulating protective immune responses and integrity of the epithelial layer [11,12,13,14]. 

TCR binding to the MHC-peptide complex is not sufficient to initiate the T cell response. Co-stimulatory signals are required, namely the binding of CD40 and of CD80/CD86 presenting on APCs to its ligand, CD40L and CD28, on T cells, respectively. T cell stimulation by cytokines produced by APCs is also important to promote the immunological synapse [1,10]. 

Upon correct activation, T cells secrete cytokines and differentiate into effector and memory T cells. Based on the profile of cytokines produced, Th cells can be classified as Th1, Th2, and Th17. Th1 cells are stimulated by interleukin (IL)-12 and produce the pro-inflammatory cytokines IL-12 and IFN-γ, participating in the cellular immune responses. Th2 cells are stimulated by IL-1 and produce the anti-inflammatory cytokines IL-4, IL-13, and IL-10, and commonly participate in the humoral immune responses. The polarization of the immune response toward a Th1 or Th2 depends on the characteristics of the immunological synapse and the type of cytokines present in the microenvironment [1,10]. 

Suppressor T cells or regulatory T cells (Tregs) are indispensable for maintaining immune tolerance and homeostasis; they can inhibit activation, proliferation, and effector functions of a wide range of immune cells [15,16,17].

B cells can recognize most antigens without previous processing through their BCR (T independent response). However, optimal response requires Th cell co-stimulation (T dependent response). Once stimulated, B cells differentiate into memory cells and plasma cells, producing immunoglobulins, initially of the IgM class and then IgG, IgA, and IgE [1,10].

## 3. Canine Pregnancy

Pregnancy length in bitches is highly variable when measured from the day of breeding, but is relatively consistent when measured from the day of ovulation (63 ± 2 days) and peak of luteinizing hormone (65 ± 2 days) [18]. During pregnancy, both embryonic and maternal factors contribute to a successful pregnancy. Canine pre-implantation embryos promote the development of specific uterine immune environment to prevent maternal immune attacks and prepare its own adhesion, nidation, and further development. For instance, embryos present CD4 molecules, but do not express the genes that encode for MHCI and MHCII molecules, which might prevent its recognition as foreign antigens by maternal cytotoxic CD8^+^ T cells [19,20]. 

Maternal thymus undergoes involution in all mammalian species examined [21], and in the mouse, at least, thymus involution is required for a successful pregnancy [22]. In this species, pregnancy-induced thymic involution is characterized by a reduction of all major thymic lymphoid and non-lymphoid cell populations [23]. Thymic cortex shrinks and medulla enlarges, increasing the number of mature thymocytes. It is suggested that the new T cell population may have suppressive/regulatory functions, and thereby contribute to the immune suppression of the female toward fetal antigens. Furthermore, the pregnancy-associated cortical involution may reflect the deletion of potentially reactive T lymphocytes, promoting immune tolerance towards fetus. The complete re-establishment of the organ occurs at the end of the lactation [15,16,17,21,24] (Figure 1A). 

Furthermore, pregnancy hormones (progesterone and 17β-estradiol) have immune modulatory properties, acting on several immune cells and influencing cytokine production [25]. Indeed, it was observed an elevation of serum concentrations of anti-inflammatory IL-4 and IL-10 during early gestation, probably induced by progesterone, a potent inhibitor of pro-inflammatory response. Between 30 and 40 days of gestation, the concentration of anti-inflammatory cytokines decreases as a consequence of high levels of prolactin, which possesses immune regulatory properties. Also, during the third phase of gestation, an increase of IL-10 was again detected, probably mediated by 17β-estradiol. The concentration of inflammatory IL-12A (IL-12p35) remained persistently low throughout canine pregnancy, pointing towards a Th2 polarization of the immune response [26]. 

During pregnancy, vaccination is generally not recommended, especially with modified live vaccines (MLV). Exceptions can be admitted, especially in shelters during a disease outbreak where vaccination can be advised [27]. Some authors refer to the risks of inducing disease in females and of infecting growing fetuses and neonates, especially with parvovirus [27,28,29]. However, a multivalent canine vaccine constituted by MLV core strains (canine distemper virus, canine adenovirus, and canine parvovirus) and canine parainfluenza virus, and inactivated *Leptospira* and rabies fractions proved to be safe for pregnant bitches and their offspring [29]. 

## 4. Maternally Derived Antibodies (MDA)

The neonatal period (from birth to 21 days of life) is of major risk, as approximately 10–30% of live-born puppies die before reaching 21 days old, mainly due to septicemia in the first three days of life [30,31]. 

The type of canine placentation (endotheliochorial) limits the transfer of immunity to the fetus. Therefore, newborn puppies are nearly agammaglobulinemic, presenting at birth IgG serum levels of 0.3 g/L, while adult dogs present 8–25 g/L [32,33,34].

Thus, passive transfer of maternally derived antibodies (MDA) through colostrum is fundamental for newborn puppy survival [35,36,37,38]. Colostrum, the specific secretion of the mammary gland produced during the first two days post-partum, is rich in immunoglobulins (Figure 1B). Intestinal Ig absorption is optimized within the first four hours, decreasing after that, until gut closure that occurs at 16–24 h after birth [39]. Intestinal barrier closure seems to be related to the maturation of intestinal cells and establishment of intestinal microflora [35]. 

Several studies have examined Ig composition of canine colostrum and milk [40,41,42,43,44,45]. IgG is the most abundant Ig isotype that can be found in the mammary secretion, remaining elevated during the first day’s post-partum. After that, IgG concentrations rapidly decreased, and IgA and IgM persist raised until weaning [41,44].

The immune quality of colostrum is highly variable among bitches and mammary glands, showing no relationship with the maternal blood IgG level, dam’s age, breed size, or litter size [46]. In addition to ensuring systemic immune defense, colostrum may also provide other immune compounds in dogs, as occurs in other species. Except for IgA that participates in the neutralization of enteric pathogens, the other nonspecific compounds of the colostrum, such as lysozyme and lactoferrin are considered of minor importance [45,47,48,49]. 

Colostrum ingestion contributes to gastrointestinal tract maturation in dogs, with the major changes occurring within the first 24 h postpartum. The colostrum ability to elicit a hypertrophic and hyperplastic response has been attributed to the presence of nutrient components, namely antibodies, hormones, and growth factors [50]. 

The duration of MDA was studied. The reported half-life for maternally transferred distemper and canine infectious hepatitis IgG was approximately eight days in puppies [35,51] and, on average, passively acquired antibodies to distemper and canine parvovirus declined to insignificant levels at about 10–12 and 15 weeks, respectively [35]. It has been suggested that dog growth rate contributes to the kinetics of MDA disappearance, with rapidly growing breeds eliminating antibodies more rapidly than slow growth breeds [35,52].

As MDA decreases, the level of maternal antibodies is no longer sufficient to prevent infections, but avoids the active immunization, creating a critical period of time (“window of susceptibility” or “immune gap”) in which the dog is susceptible to infections [35,53]. Several studies have addressed the interference of MDA in the seroconversion after vaccination against highly pathogenic microorganisms, namely canine parvovirus [54,55,56] and canine distemper virus [57].

In parallel to MDA decreasing, and in response to environmental pathogens, the newborn puppy produces its own antibodies, with a significant increase of antibody concentration as early as 14–21 days of age [58].

## 5. Development and Maturation of Lymphoid Organs

The ontogeny of the canine immune organs was reviewed in a few publications [36,37,59]. Hematopoietic and immune cells arise from a common bone marrow stem cell. Thereafter, B cells undergo maturation in the fetal liver and bone marrow, which represent successive primary lymphoid organs. B cells maturation involves the acquisition of BCR and selection to ensure that only B cells that express functional BCR (positive selection) and do not ligate self-antigens (negative selection) survive. On the other hand, immature T cells are exported to the thymus for final maturation [1,60,61]. The thymus generates a diverse repertoire of T cells that undergo positive and negative selection, ensuring that autoreactive cells are eliminated before reach peripheral organs [1,62]. Monocytes and granulocytes (neutrophils, eosinophils, and basophils) mature in the bone marrow and are released into the bloodstream [1].

Maturation of the immune system occurs from birth to approximately six months old. Although the puppy was considered immunocompetent between 6–12 weeks of age, it is not possible to predict accurately the onset of immunocompetence, since it depends on the presence of MDA [38].

In growing animals, hematopoietic bone marrow is located inside the long and flat bones, but as the animal ages the medullary cavity is replaced by fatty tissue and active bone marrow is confined to the trabecular cavities of flat bones, and epiphyses and metaphysis of long bones [63]. The proportion of T helper cells (CD4^+^), cytotoxic T cells (CD8^+^), and ‘unconventional’ γδ-T cells increases with age. Mature T cells (CD3^+^) reaches more than 60% in adult dogs (Figure 2). With increasing age, bone marrow plays a dual role of primary and secondary lymphatic organ, maintaining a pool of effector and memory lymphocytes [64].

Thymus rapidly grows in dogs, reaching maximum size at six months of age. Then, when the dog reaches sexual maturity (between 6 and 23 months), the organ suffers involution, characterized by reduction of thymic parenchyma, which is replaced by adipose, connective tissue, and prominent epithelial structures (cords, tubules, cysts) [24,65,66,67]. In newborn puppies, the organ comprises approximately 12% of CD4^+^ and 3% of CD8^+^ T cells, of which 69% were double-positive (CD4^+^CD8^+^) and 13% double negative (CD4^−^CD8^−^) cells [68]. The CD4^+^CD8^+^ double positive phenotype characterizes immature T cells during thymic development [69]. Approximately 5% of the cells recovered by thymus tissue teasing were CD34^+^ progenitor cells [70].

Secondary lymphoid tissues include the encapsulated organs (lymph nodes and spleen) and mucosal associated lymphoid tissue (MALT). Secondary lymphoid tissues facilitate interactions between naïve lymphocytes and APC, leading to productive adaptive immune responses [71].

The distribution of lymphocyte subsets in secondary lymphoid tissue of one-day-old puppies and adult dogs was studied. The percentage of T cells (CD3^+^) was lower in the spleen compared with lymph nodes. Conversely, B cells (CD21^+^) predominate in the spleen relative to other compartments (Figure 2). The highest proportion of B cells observed in the spleen of puppies may be related to its function as primary B-lymphopoietic organ. The subsequent reduction of B cell population in adult dogs may result from the peripheral negative selection in the spleen and recirculation of these cells to the induction sites, namely lymph nodes. A relatively high number of γδ-T cells were found in the spleen, although very few cells were present in lymph nodes [72]. These cells retain cytotoxic activity and constitute a first line of defense of epidermal and mucosal epithelial linings [14,73].

MALT comprises non-encapsulated lymphoid tissue that is continuously exposed to antigens against which it is necessary to mount an immune response or maintain immune tolerance [76,77,78,79]. The presence of nasal associated lymphoid tissue was documented in dogs [67], but bronchus associated lymphoid tissue seems not to be a constitutive structure [77]. Gut associated lymphoid tissue (GALT) is characterized by agglomerates of lymphocytes, organized into discrete follicles denominated Peyer’s patches within the mucosa, as well as lymphocytes scattered throughout the lamina propria [67]. Dogs have two distinct types of Peyer’s patches: duodenal and jejunal (proximal units), and single ileal Peyer’s patches. The proximal units contribute to the mucosal immune response, while the ileal Peyer’s patches present features of a primary lymphoid organ, participating in the early development of the B cell system, and show involution when dogs reach sexual maturity [80,81]. There is a remarkable increase in the weight of mesenteric lymph nodes after weaning, which reflects the importance played by lymph nodes in fighting exogenous antigens [65]. 

Regarding circulating leukocyte population, polymorphonuclear neutrophils (PMN) predominates in the first day of life and were almost three times higher than lymphocyte count. During the first week, there is a decrease in PMN and a transient predominance of lymphocytes, which might reflect immune system activation through contact with foreign antigens [74]. Reference values of blood cells determined for puppies with age ranging between 16 and 60 days old are presented in Table 1.

The phenotype of the circulating lymphocyte subpopulation in neonatal dogs differs significantly from that of adult dogs. Indeed, peripheral blood of a one-day-old puppy contains a lower proportion of T cells, with a very low rate of CD8^+^ T cells, and a high amount of B cells [72,74]. Although the percentage of blood CD4^+^ T cells remains relatively stable from birth to adulthood, the amount of CD8^+^ T cells increases with age. The high levels of circulating B cells observed in newborn puppies decreased progressively with age. Moreover, it was also reported a nonsignificant age-related decrease of γδ-T lymphocytes [74,75] (Figure 2).

Thus, newborn puppies present a developing immune system, which is different from adults. The fetus lives in the uterus, which is a sterile environment, ensuring that T and B cells of the newborn not met its cognate antigen in the periphery (naïve cells) and that activated, and memory cells were absent [83]. After birth, the newborn is exposed to microbial-rich surroundings. The exposure to external antigens stimulates the immune system, inducing the massive activation and redistribution of peripheral lymphocytes, which change the size and structure of lymphoid organs and promote the appearance of lymphocytes in previously empty spaces [72]. The maturation of the immune system progresses from birth to approximately six months old [38].

## 6. Development and Maturation of the Immune Response

Ontogeny studies of the immune response revealed that the fetus produces a specific antibody response to T cell-dependent antigens (bacteriophage ØX-174), ovine erythrocytes, and *Brucella canis*. Fetal T cells of the spleen, lymph nodes, and thymus respond to the mitogen phytohemagglutinin (PHA). These studies showed that the fetus possesses a functional lymphocyte (B and T cells) system able to generate humoral and cellular immune responses against several antigens, suggesting that newborns are immunocompetent close to, or at birth [36,37,38,72]. 

Furthermore, some studies confirmed that colostrum-deprived one-day-old puppies are able to mount humoral immune responses to some antigens [35,84]. Besides, it was demonstrated that puppies with different ages develop lymphoproliferative responses to several types of mitogen [72,74,85], contradicting the idea of antigen neonatal tolerance.

However, MDA inhibits the development of endogenous neonatal immune response and constitutes the main obstacle to successful vaccination [35,38,53]. In an attempt to determine the optimal age to initiate vaccination, the specific humoral immune response elicited by several vaccine preparations was studied in puppies with different ages and from distinct breeds [54,74,86]. It seems that antibody response to vaccination is specific to each animal and depends on the age of the dog, protective antibody titer, and vaccine type [86]. Thus, it is assumed that around 6–12 weeks, MDA no longer interferes with the development of an adequate immune response, and puppies are considered immunocompetent (Figure 1C) [38]. Accordingly, international guidelines recommend starting vaccination at 6–8 weeks of age, then re-vaccinate each 2–4 weeks until 16 weeks of age or older, to ensure that at least one vaccine dose induces immunity [27].

In puppies, MHCII is constitutively expressed by APCs [87]. Thus, the phagocytic activity of peripheral blood leukocytes, (PMN and monocytes) at birth and in two-month-old puppies seems not to be compromised [74]. Studies realized in mice and humans have demonstrated some differences between immune responses in adults and neonates. Newborn APCs have a reduced capacity to express CD86 and CD40, and the respective ligands (CD28 and CD40L, respectively) on lymphocytes are also reduced. Binding of antigen to BCR does not induce the hyper-expression of MHC class II molecules, increase the expression of B7.2 (CD86) costimulatory molecule, or the upregulation of CD40 and CD40L. Thus, the defective interaction between B and T cells results in T cell anergy, deviation towards Th2 response, hampers specific B cell response and B cell switch to different B cell classes and subclasses [53,88,89]. Although it is unknown if the same happens in the dog, some studies point towards a Th2 polarization of the neonatal immune response. As the dog grows, the immune system undergoes an educational process provided by exposure to Th1 antigens, achieving a balanced Th1-Th2 immune response [53,90,91] (Figure 1D). However, the improvement of dog’s living conditions, good nutrition, vaccination, and deworming programs probably decreases the exposition to Th1 stimulus. Vaccination may also have a profound and long-lasting effect in driving the immune system to a Th1 or Th2 response, that was not yet investigated [90]. 

## 7. Immunosenescense

Increased life span allowed the recognition of age-related higher susceptibility to infectious, inflammatory, autoimmune, and neoplastic diseases [92]. The gradual deterioration of the immune system function associated with aging is called immunosenescense [93,94,95]. This phenomenon can result from an intrinsic ageing process related to thymus involution and decreased output of naïve cells from bone marrow and thymus (Figure 1E). However, the depletion of the reservoir of naïve cells over time by contact with pathogens and their conversion into memory cells during adaptive immune responses may also contribute to the decline of the immune function [96]. 

Despite contradictory results, some findings were consistently associated with canine elderly, namely the reduction of blood CD4^+^ T cells, expansion of the CD8^+^ cell subset with a subsequent reduction of CD4:CD8 ratio, and a decrease of naïve lymphocytes [97,98,99,100,101,102,103].

Age-related changes include impairment of the cell-mediated immune response, as demonstrated by the reduction of proliferative response of blood lymphocytes to mitogens and the reduction of cutaneous delayed type hypersensitivity. Moreover, there is a decline in the humoral immune response probably related to the decreased functionality of Th cells. The ability to mount humoral immune responses seems to prevail, as demonstrated by the persistence of protective vaccine antibody titers, and respond to booster vaccination with elevation in titer [94]. Although the currently adopted triennial re-vaccination program, instead of the prior annual re-vaccination, offers adequate protection to young and adult dogs [104], this vaccination scheme may not confer protection to geriatric dogs [105]. 

Older dogs commonly present an impairment of immune responses to novel antigenic challenges, such as infections and vaccines, which probably is related to the reduction of the peripheral pool of naïve T cells and low diversity of the repertoire of T cell receptors [95]. Indeed, first-time rabies vaccination of older dogs revealed a significant decrease in antibody titers and an increase of vaccination failure, suggesting that the elderly can compromise the primary response to vaccination [106].

Few investigations have studied the effect of cumulative antigenic exposure and the onset of late-life inflammatory disease (inflammageing) in this species. Available data suggest that elderly dogs exhibit an elevation of serum concentration of oxidative damage biomarker, indicating that they have a reduced ability to respond to oxidative stress. The implementation of strategies to slow the pro-inflammatory state, such as supplementation with antioxidants and the optimization of vaccination protocols, seems to be essential to promote a long and healthy elderly [107]. 

## 8. Concluding Remarks and Future Trends

Many aspects of the canine immune system remain unknown, namely the characterization of the innate immune response and of the contribution of inflammageing in the development of inflammatory, autoimmune, and neoplastic diseases in elderly dogs. Furthermore, the effect of improving living conditions and regular vaccination on the activity of the dog’s immune system is unknown. Knowing the functional capacity of the immune system in different phases of life can help veterinarians to delineate prophylactic and therapeutic approaches to improve dog health and longevity. It is particularly important to study the neonatal and geriatric immune system and its capacity to respond to the diversity of antigens that threaten the life of newborn and elderly dogs, respectively. In addition, dogs represent an important large animal model to humans, developing similar immune, neoplastic, infectious, and parasitic diseases, which allow the study of many immune aspects in a short period. For example, the similarities between human and dog’s immunosenescence makes the dog an important model of study, allowing the development of strategies to improve life, especially during elderly.

## Figures and Tables

**Figure 1 vetsci-06-00083-f001:**
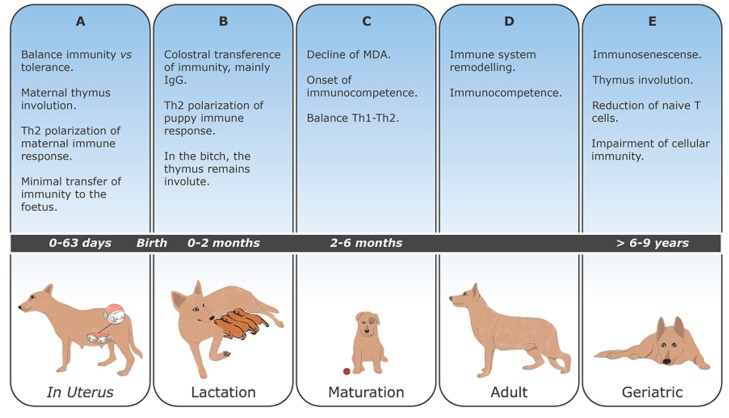
Major changes in the dog´s immune system lifelong (**A**–**E**).

**Figure 2 vetsci-06-00083-f002:**
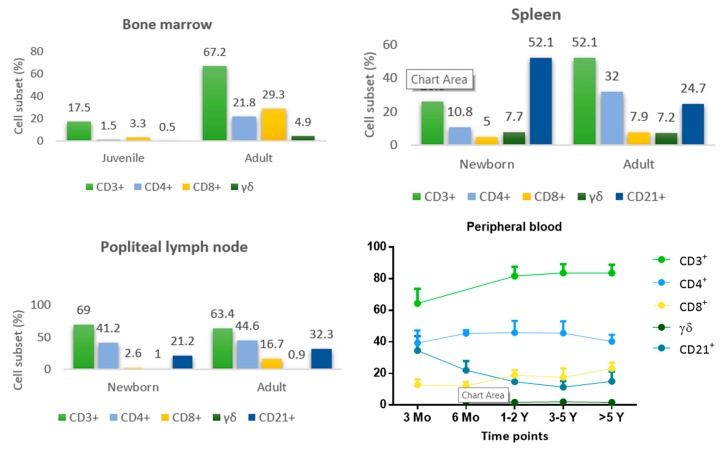
Lymphocyte subset distribution in bone marrow, spleen, and popliteal lymph node of newborn puppies and adult dogs [64,72], and lymphocyte subsets in peripheral blood of Beagle dogs determined at different ages [74,75].

**Table 1 vetsci-06-00083-t001:** Hematologic parameters of puppies with age ranging between 16 and 60 days are presented as mean (SD), as well as the reference range values of adult dogs [82].

	16–24 Days	28–46 Days	46–60 Days	Adult (Reference Range)
**WBC (×10^9^/L)**	11.0 (2.9)	13.1 (2.7)	13.7 (3.8)	6–18
**Neutrophils (×10^9^/L)**	5.5 (2.4)	7.0 (1.8)	7.1 (2.1)	3.6–13.0
**Lymphocytes (×10^9^/L)**	4.2 (1.4)	4.6 (1.1)	5.0 (2.1)	0.8–5.8
**Monocytes (×10^9^/L)**	0.6 (0.18)	0.9 (0.2)	0.9 (0.3)	0–1.6
**Eosinophils (×10^9^/L)**	0.5 (0.26)	0.4 (0.25)	0.5 (0.39)	0–1.8

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
