# Peer review of "Development of Dog Immune System: From in Uterus to Elderly"

_vetsci, 2019, doi:10.3390/vetsci6040083_

Round 1
Reviewer 1 Report
This review is well organized and comprehensive. It also includes multiple references to the pertinent literature and informative figure that adequately accompany what reported in the text. I have only some suggestions.
Reviewer comments on the manuscript “vetsci-586034 entitled ‘Development of dog immune system: from in uterus to elderly”
Canine pregnancy
Line 39-41: I do not well understand the meaning of this paragraph: please explain better the concept of thymus involution and rearrangement of cellular microenvironment.
Line 40-41: References in veterinary medicine, especially in dogs regarding the involvement of “T cells with suppressive functions” (T regulatory cells) should be included to support the concept “the generation of T cells with suppressive functions in dogs”.
Maturation of lymphoid organs
Line 139-140: I do not well understand the meaning of this period: please delete the word “existence” because “naïve” cells are present in newborn as well as in adult.
Maturation of the immune response
Line 167: “B7.2” and “(CD40L)” is not the same. Please correct the sentence.
Line 163-167: References are available in dogs?
Line 168: T-cell energy? Probably you meant T-cell anergy. If so, please correct.
Immunosenescence
Line 198: You write “ ….a reduced ability to respond to intracellular oxidative stress”. This statement concerns T Lymphocytes or the immune system in general? Please clarify better.
Line 190: You could better explain the involvement of the humoral immune response, for example: insert after “hypersensitivity” the following sentence: “Moreover, there is a decline in the humoral immune response probably related to the decreased functionality of T helper lymphocytes”.
Line 192: Please replace “However “with another word.
Author Response
Reviewer comments on the manuscript “vetsci-586034 entitled ‘Development of dog immune system: from in uterus to elderly”
The authors are very grateful to the editor and reviewers for the constructive comments and suggestions raised.
Reviewer 1
Canine pregnancy
Line 39-41: I do not well understand the meaning of this paragraph: please explain better the concept of thymus involution and rearrangement of cellular microenvironment.
Thank you very much for your comment. The concept of thymus involution and rearrangement of cellular microenvironment was better explained (lines 127-135).
Line 40-41: References in veterinary medicine, especially in dogs regarding the involvement of “T cells with suppressive functions” (T regulatory cells) should be included to support the concept “the generation of T cells with suppressive functions in dogs”.
References on dog T regulatory cells were added (lines 133-135).
Maturation of lymphoid organs
Line 139-140: I do not well understand the meaning of this period: please delete the word “existence” because “naïve” cells are present in newborn as well as in adult.
The mentioned period was rewritten taking in to account the above comments (lines 275-282).
Maturation of the immune response
Line 167: “B7.2” and “(CD40L)” is not the same. Please correct the sentence.
The sentence was corrected. Thank you very much for noticing (line 310).
Line 163-167: References are available in dogs?
Despite we have done an extensive search in the available literature, we did not find anything referring to antigen presentation in newborn puppies.
Line 168: T-cell energy? Probably you meant T-cell anergy. If so, please correct.
The correction was made (line 312). Thanks.
Immunosenescence
Line 198: You write “ ….a reduced ability to respond to intracellular oxidative stress”. This statement concerns T Lymphocytes or the immune system in general? Please clarify better.
The statement was rewritten taking into account the reviewers suggestion (lines 341-343).
Line 190: You could better explain the involvement of the humoral immune response, for example: insert after “hypersensitivity” the following sentence: “Moreover, there is a decline in the humoral immune response probably related to the decreased functionality of T helper lymphocytes”.
Thank you very much for the suggestion. The suggested sentence was included (lines 333-334).
Line 192: Please replace “However “with another word.
“However” was replaced by “Still” (line 336). Thank you for noticing.
Reviewer 2 Report
Comments to authors:
The review aimed to summarize the different stages of immune system development in the dog. Changes in the immune system during the elderly were also mentioned. The topic is interesting and this kind of reviews are needed in the canine medicine.
However, the here presented study is unacceptable for a publication in the actual form due to few reasons:
1) There is no introduction to the subject: why this work was done? What is the objective of this review? How the subject will be presented?
2) The article is too synthetic. It gives an idea of the subject but at the academic, and not scientific level. For instance, many cited data merit to be developed (like mentioned below kinetics of maternally derived antibodies, or the weight of different immune organs, etc.). On the other site, authors used many abbreviations and speak about some details of immune response mechanism, things which are impossible to be understood by a non-specialist in immunology. Therefore, I would suggest to either go dipper in the mechanistic part of the developmental immunology in the dog, or give a good basic information for a non-specialized reader (including function of different lymphocyte subsets, but also the general white blood cells profile since birth until aging).
3) The authors make statements based on studies taken form the human medicine; however, the reader should be able to know what has been demonstrated in the dog, and what is just an extrapolation.
4) As I am not a native speaker in English, I would suggest a revision of the English by a native speaker. I have an impression that some sentences are not built correctly (line 143-144).
Specific comments:
Line 21 – “in the uterus “ should be removed as the passive transfer in the uterus in a minor source of the passive immunity in the dog (only about 5-10%).
Line 29 – I would add “Passive immune transfer”, “Immunity development”,
Line 32 – it is exactly 63±2 days (since the ovulation date) or 65±2 days (since the peak of LH).
Line 43 – Fig.1 Although the figure is very clear and well summarizing the changes all along the dog’s life, its quality is very poor, making almost impossible to read the text. Please provide a figure of a better quality.
Line 44 – if the changes last during the lactation period this information should be also included in the Fig.1.
Line 54-57 – the sentence is too long and not easy to understand. What exactly was influenced by the high level of progesterone? The offspring at which age (at birth which would directly reflect the fetal life or later)? Was the offspring controlled for Toxocara infestation, as we know that a vast majority of dogs are born positive to Toxocara due to the transplacental transfer. And then, is it due to the hormonal influence of the parasite per se that the immune response is altered? Finally, how is it possible to make any observations on puppies in an in vitro study?
Line 61 – The pregnancy in the dog last 2 months, then we cannot speak about a trimester. Was the statement that the passive immune transfer occurs during the last part of pregnancy demonstrated in the dog? If so, could you provide a reference? And if not, could you state that this data is obtained for humans.
Line 63 – Fig.1 does not present the concentration of IgG in the colostrum.
Line 68 – 72 hours after …. progressively decreases – do you mean between parturition and 72h or 72h and 6 weeks? It is not clear in the sentence.
Line 75 – the reader has an impression that these data were obtained in the dog, while a review on the dog is used as a reference, in which rather examples from the human medicine are stated. Please provide more details, including origins from these data.
Line 76 – How does the non-specific and specific immunity decreases? Is there any difference in the kinetics depending on the breed, environment, etc.? The ‘immune gap”, is it a general problem or it concerns only some diseases?
Fig.2 – Please, provide the raw values of lymphocytes proportions.
Line 123 – Due you mean an increase in size? This should be precise.
Fig.3 – Is it “unpublished data” from the authors? If not, a reference should be mention in the figure title. Why does the total number of different lymphocytes for each time point is above 100%?
Line 164; 166 – Please explain the abbreviations.
Author Response
Reviewer comments on the manuscript “vetsci-586034 entitled ‘Development of dog immune system: from in uterus to elderly”
The authors are very grateful to the editor and reviewers for the constructive comments and suggestions raised
Reviewer 2
The review aimed to summarize the different stages of immune system development in the dog. Changes in the immune system during the elderly were also mentioned. The topic is interesting and this kind of reviews are needed in the canine medicine.
However, the here presented study is unacceptable for a publication in the actual form due to few reasons:
1) There is no introduction to the subject: why this work was done? What is the objective of this review? How the subject will be presented?
Taking into account the reviewer comment, an introduction section was added answering the raised questions by the reviewer (lines 32-45).
2) The article is too synthetic. It gives an idea of the subject but at the academic, and not scientific level. For instance, many cited data merit to be developed (like mentioned below kinetics of maternally derived antibodies, or the weight of different immune organs, etc.). On the other site, authors used many abbreviations and speak about some details of immune response mechanism, things which are impossible to be understood by a non-specialist in immunology. Therefore, I would suggest to either go dipper in the mechanistic part of the developmental immunology in the dog, or give a good basic information for a non-specialized reader (including function of different lymphocyte subsets, but also the general white blood cells profile since birth until aging).
Taking into accounting the reviewer comment, the authors added a basic description of constitution and function of mammal immune system (Immune system overview - lines 47-117).
However, the authors agreed with the reviewer comment that it is necessary to deepen many of the immune mechanisms associated with dog development. Nevertheless, the available information related to many of the aspects of immune development and immune response are scarce, pointing towards the need for additional studies able to clarify many of these issues.
3) The authors make statements based on studies taken form the human medicine; however, the reader should be able to know what has been demonstrated in the dog, and what is just an extrapolation.
Authors acknowledge reviewer comment, and the indication of species where studies were carried out was added to the manuscript namely in what concerns thymus involution during pregnancy (lines 127-135), Th2 polarization of the immune response (lines 147-151), transfer of others immune compounds by colostrum (lines 172-179).
4) As I am not a native speaker in English, I would suggest a revision of the English by a native speaker. I have an impression that some sentences are not built correctly (line 143-144).
The indicate sentence was improved (lines 275-282) and a revision of the entire manuscript was done.
Specific comments:
Line 21 – “in the uterus “should be removed as the passive transfer in the uterus in a minor source of the passive immunity in the dog (only about 5-10%).
“in the uterus” was removed as suggested (line 21).
Line 29 – I would add “Passive immune transfer”, “Immunity development”.
The suggested key words were added. Many thanks for the suggestions (line 29).
Line 32 – it is exactly 63±2 days (since the ovulation date) or 65±2 days (since the peak of LH).
Sentence about pregnancy length has been improved, now referring pregnancy duration in relation to LH peak and ovulation (lines 119-121).
Line 43 – Fig.1 Although the figure is very clear and well summarizing the changes all along the dog’s life, its quality is very poor, making almost impossible to read the text. Please provide a figure of a better quality.
The quality of Figure 1 was improved.
Line 44 – if the changes last during the lactation period this information should be also included in the Fig.1.
This information was added to Figure 1.
Line 54-57 – the sentence is too long and not easy to understand. What exactly was influenced by the high level of progesterone? The offspring at which age (at birth which would directly reflect the fetal life or later)? Was the offspring controlled for Toxocara infestation, as we know that a vast majority of dogs are born positive to Toxocara due to the transplacental transfer. And then, is it due to the hormonal influence of the parasite per se that the immune response is altered? Finally, how is it possible to make any observations on puppies in an in vitro study?
The reviewer is right. The authors recognize that the reference indicated in the manuscript is not the most appropriate to demonstrate the influence of the fetal-placental environment on the polarization of the newborn's immune response. Therefore, the sentence was deleted, but the idea was maintained stating that studies were conducted in humans and rats. (lines 147-151).
Line 61 – The pregnancy in the dog last 2 months, then we cannot speak about a trimester. Was the statement that the passive immune transfer occurs during the last part of pregnancy demonstrated in the dog? If so, could you provide a reference? And if not, could you state that this data is obtained for humans.
The statement was changed (lines 158-160) and references demonstrating transplacental transport of IgG in bitches and the serum levels of IgG in newborns at birth were included (176-185).
Line 63 – Fig.1 does not present the concentration of IgG in the colostrum.
This information was added to figure 1.
Line 68 – 72 hours after …. progressively decreases – do you mean between parturition and 72h or 72h and 6 weeks? It is not clear in the sentence.
The sentence was improved (lines 167-167).
Line 75 – the reader has an impression that these data were obtained in the dog, while a review on the dog is used as a reference, in which rather examples from the human medicine are stated. Please provide more details, including origins from these data.
The text was changed, indicating data origin, and suitable citations were included (lines 172-175).
Line 76 – How does the non-specific and specific immunity decreases? Is there any difference in the kinetics depending on the breed, environment, etc.? The ‘immune gap”, is it a general problem or it concerns only some diseases?
Although we have made a comprehensive literature search, data on the immune kinetics related to dog development or dogs breed, and environment are scarce. Even so, were included new data on innate immunity, MDA in fast-growing breeds and virus disease potential lethal (lines 180-185, 187-193, 304-306).
Fig.2 – Please, provide the raw values of lymphocytes proportions.
Figure 2 was changed taking into account the reviewer comment and the values of lymphocyte populations were included.
Line 123 – Due you mean an increase in size? This should be precise.
The sentence was rephrased indicated the weight increase of mesenteric lymph nodes (line 261).
Fig.3 – Is it “unpublished data” from the authors? If not, a reference should be mention in the figure title. Why does the total number of different lymphocytes for each time point is above 100%?
The references were added, and the standard deviation was introduced. All T lymphocyte subsets are CD3+. For instances, Th cells are CD3+CD4+ and cytotoxic T cells are CD3+CD8+.
Line 164; 166 – Please explain the abbreviations.
The abbreviations used are explained in the beginning of the manuscript under the subtitle Immune system overview.
Round 2
Reviewer 2 Report
The quality of the manuscript was strongly improved. Although, some issues remain to be addressed.
As this review is destined for veterinary practitioners, I would suggest to provide readers with a synthetic table of specific blood immune parameters (i.e. white blood cells count) at different periods of life (if possible at some key timepoints). What is more, I would expect to present a more specific comments on the consequences of such immune system evolution on the dog health. Until now, only the response to the vaccination in the growing dog was commented. What about the vaccination of the old dog or pregnant female dog? Is it possible to underline any other consequences of such changes for the veterinarian?
Is it possible to provide reader with more explanation about γδ? For instance, what is the role of γδ as it seems to be increased in the bone marrow and in the peripheral blood in the adult dog but not in the newborn?
Specific comments:
Line 127 - sentence to be reformulated
Line 128 - "In this species" Which species do authors mean?
Line 148-151 - The sentence is too long and difficult to follow. What kind of fetal-placental tissues do authors mean? And how can fetal-placental tissues immune status influence fetus/newborn?
Line 167-169 - The kinetics of immunoglobulins' cconcentrations in the colostrum should be verified as few studies describe other evolution pattern (Albaret et al., 2016; Schafer-Somi et al., 2005; Reynolds and Johnston 1970).
Line 289 -The sentence needs to be reformulated.
Line 305 - The sentence must be revised, for example "... in newborn and in two-month-old puppies..." would be more suitable.
Author Response
The authors appreciate the careful and interesting manner in which Reviewer 2 approached this review. Comments and suggestions raised have greatly contributed to manuscript improving.
As this review is destined for veterinary practitioners, I would suggest to provide readers with a synthetic table of specific blood immune parameters (i.e. white blood cells count) at different periods of life (if possible at some key timepoints). What is more, I would expect to present a more specific comments on the consequences of such immune system evolution on the dog health. Until now, only the response to the vaccination in the growing dog was commented. What about the vaccination of the old dog or pregnant female dog? Is it possible to underline any other consequences of such changes for the veterinarian?
A: All indicated concerns were addressed in the text (lines 149-155; 346-355; 369-376)
Is it possible to provide reader with more explanation about γδ? For instance, what is the role of γδ as it seems to be increased in the bone marrow and in the peripheral blood in the adult dog but not in the newborn?
A: Some more information on gamma delta lymphocytes has been added, although its function in the dog's spleen is unknown (lines 98-99, 235-237).
Specific comments:
Line 127 - sentence to be reformulated
A: The sentence was rephrased (line 128-129).
Line 128 - "In this species" Which species do authors mean?
A: The sentence was improved (line 128-129).
Line 148-151 - The sentence is too long and difficult to follow. What kind of fetal-placental tissues do authors mean? And how can fetal-placental tissues immune status influence fetus/newborn?
A: The previous sentence was improved and the mentionated sentence was removed.
Line 167-169 - The kinetics of immunoglobulins' cconcentrations in the colostrum should be verified as few studies describe other evolution pattern (Albaret et al., 2016; Schafer-Somi et al., 2005; Reynolds and Johnston 1970).
A: Ig concentration was reformulated, taking into account different reports (lines 171-173).
Line 289 -The sentence needs to be reformulated.
A: The sentence was rephrased (lines 286-293).
Line 305 - The sentence must be revised, for example "... in newborn and in two-month-old puppies..." would be more suitable.
A: The reviewer’s suggestion was taken into consideration, and the assessment was improved (lines 315-317).
